

# Evolution of *Helicobacter* spp: variability of virulence factors and their relationship to pathogenicity

Carlos F. Prada[1,2], Maria A. Casadiego[2] and Caio CM Freire[1]

[1] Department of Genetics and Evolution, Federal University of Sao Carlos, Sao Carlos, Sao Paulo, Brazil
[2] Grupo de Investigación de Biología y Ecología de Artrópodos. Facultad de Ciencias., Universidad del Tolima, Tolima, Colombia

Corresponding author
Caio CM Freire, caiofreire@ufscar.br

## ABSTRACT

**Background:** Virulence factors (VF) are bacteria-associated molecules that assist to colonize the host at the cellular level. Bacterial virulence is highly dynamic and specific pathogens have a broad array of VFs. The genus *Helicobacter* is gram-negative, microaerobic, flagellated, and mucus-inhabiting bacteria associated with gastrointestinal inflammation. To investigate about their pathogenicity, several *Helicobacter* species have been characterized and sequenced. Since the variability and possible origin of VF in the genus are not clear, our goal was to perform a comparative analysis of *Helicobacter* species in order to investigate VF variability and their evolutionary origin.

**Methods:** The complete genomes of 22 *Helicobacter* species available in NCBI were analyzed, using computational tools. We identified gain and loss events in VF genes, which were categorized in seven functional groups to determine their most parsimonious evolutionary origin. After verifying the annotation of all VF genes, a phylogeny from conserved VF organized by *Helicobacter* species according to gastric *Helicobacter* species (GHS) or enterohepatic (EHS) classification was obtained.

**Results:** Gain and loss analysis of VF orthologous in *Helicobacter* ssp revealed the most possible evolutionary origin for each gene set. Microevolutionary events in urease and flagella genes were detected during the evolution of the genus. Our results pointed that acquisition of ureases and adherence genes and deletion of cytotoxins in some lineages, as well as variation in VF genes copy number, would be related to host adaptation during evolution of the *Helicobacter* genus. Our findings provided new insights about the genetic differences between GHS and EHS and their relationship with pathogenicity.

# INTRODUCTION

The *Helicobacter* genus are Gram-negative, microaerobic, spiral-shaped bacterial species, flagellated, mucus-inhabiting bacteria associated with gastrointestinal inflammation and, in species such as *H. pylori*, the development of cancer (*Mannion, Shen & Fox, 2018*; *Smet et al., 2018*). Almost 49 *Helicobacter* spp. have been reported (https://www.ncbi.nlm.nih.

gov/genome/browse/#!/overview/Helicobacter), and isolated from stomach, gastrointestinal tract, liver, and gallbladder in more than 142 vertebrate species of mammals, birds, and reptiles (*Mannion, Shen & Fox, 2018*; *Schrenzel et al., 2010*). These bacteria are classified as gastric *Helicobacter* species (GHS) and enterohepatic *Helicobacter* species (EHS); they differ in several aspects, *e.g.*, the tropism for different parts of the gastrointestinal system and the phylogenetic relationships. GHS have been found to colonize the stomach of humans, dogs, cats, cheetahs, rhesus monkeys, ferrets, sheep, cattle, whales, and dolphins, while EHS is found in the liver, gall bladder or gastrointestinal tract, isolated in animals such as mice, rats, and hamsters (*Fox, 2002*; *Gueneau & Loiseaux-De Goër, 2002*). Although the focus is on *H. pylori* (one of the most important human pathogens, infecting approximately half of the global population) and related GHS, the mechanisms of colonization and potential virulence from EHS are interesting. Therefore, we focused our analyses on molecular features mainly on the genes associated with the formation of gastric cancer, such as the *vac* and *cag* genes (*Mannion, Shen & Fox, 2018*; *Palframan, Kwok & Gabriel, 2012*; *Schrenzel et al., 2010*). Although genes such as flagella genes have been studied mainly in *H. pylori*, indicating that these genes are of great importance in the pathogenicity (*Duan et al., 2013*; *Tsang & Hoover, 2015*), a comparative study about these genes from other species of the *Helicobacter* genus has not been performed.

In order to infect and survive the harsh milieu of mammalian gastrointestinal tract, both GHS and EHS had to adapt with various virulence factor (VF) genes. While several pathogenic mechanisms have been identified and characterized in *H. pylori*, these are less understood for other gastric species and EHS (*Mannion, Shen & Fox, 2018*; *Smet et al., 2018*). For example, virulence of *H. pylori*, as others GHS, depends on its capacity to persistently colonize the stomach, a hostile and acidic niche (*Vinella et al., 2015*). In addition, the significance of VF genes extends beyond the survival needs of the bacteria, making *H. pylori* one of the most well-adapted human pathogens, capable of sustaining a persistent infection for long periods of time (*Šterbenc et al., 2019*). The specialization of GHS to colonization of the gastric environment and the duplication at the origin of *hpn-2* (nickel-binding proteins gene) occurred in the common ancestor of *H. pylori* and *H. acinonychis* (*Vinella et al., 2015*), which demonstrates the impact on gain and loss of genetic information with adaptive processes in these pathogens.

Due to the recent ability to isolate and elucidate the complete genome of pathogenic bacteria, information from hundreds of *H. pylori* strains are now available as well as 49 complete and partial genomes of *Helicobacter* species (https://www.ncbi.nlm.nih.gov/genome/browse#!/overview/helicobacter). From these genomes, several comparative genomic analyses have been carried out (*Cao et al., 2016*; *Mannion, Shen & Fox, 2018*; *Smet et al., 2018*; *Vinella et al., 2015*) to investigate features of pathogenicity. Moreover, genetic studies have identified about 90 VFs in *H. pylori* strains (*Cao et al., 2016*; *Javed, Skoog & Solnick, 2019*; *Šterbenc et al., 2019*). However, there is no detailed comparative genomic study about the variability and evolution of virulence factors in *Helicobacter* species. Therefore, this study aimed to investigate VFs in the complete genomes of the *Helicobacter* genus with focus on genetic variability and pathogenic phenotypes.

## MATERIALS AND METHODS

### Complete genome sequence collection

The complete genomes sequences belonging to the *Helicobacter* genus were downloaded from the NCBI database (https://www.ncbi.nlm.nih.gov/genome/?term=) as of December 20, 2020. *Helicobacter* spp. genomes with incomplete sequences in scaffold or contigs phase were not taken into account in order to avoid false negatives in the genetic analyses. Previous comparative analyses (results not shown in this work) showed that *Campylobacter fetus* (NC_008599) is a high-quality genome assembly/annotation genome compared to *C. jejuni* (NC_002163.1) with 1,731 CDS *vs* 1,572 CDS, respectively. In order to avoid false negatives *C. fetus* was taken as outgroup during our analyses. Information about the *Helicobacter* spp. genomes that we investigated in listed, were summarized (Table S1).

### Identification and genetic annotation confirmation of virulence factors (VF)

A search in the different scientific databases and articles was carried out, taking as reference the list of "Virulence factors database (VFDB)" (*Liu et al., 2019*); selecting 89 genes associated to VFs described for *Helicobacter pylori* ATCC 26695 strain (NC_000915.1) that were related to the pathogenic phenotype sequence used as reference in the analyses. Similarly, they were classified into seven groups according to the metabolic function in the bacteria, taking into account databases and literature (*Javed, Skoog & Solnick, 2019*; *Liu et al., 2019*; *Sayers et al., 2019*) (Table S1).

In order to confirm the presence, orientation and location in coordinates for each VF by genome, nucleotide sequences of the 89 virulence factors were downloaded from the *H. pylori* (strain ATCC 26695) genome, using as a reference. Using as a reference both nucleotide and amino acid sequences of each of the 89 VFs of *H. pylori*, a basic local alignment search (BLAST) (*Johnson et al., 2008*) was performed on each genome analyzed; to corroborate the presence (matrix of presence and/or absence; 0, 1), position of each gene (coordinate and position matrix, plus/plus or plus/minus). Thresholds for accurate prediction of the presence of a VF were estimated with a e-value less than 0.01 and ≥85% of identity and ≥70% of coverage (*Sangar et al., 2007*). Moreover, the Geneious program (*Kearse et al., 2012*) was used to perform MUSCLE multiple alignments (*Edgar, 2004*), extracting the sequences with the exact coordinates of VF location, verifying the presence, position and size in base pairs (bp) of each gene. In all cases, gene annotation were confirmed by blast and multiple alignments with paralogous genes.

The presence of genetic gains or duplicate genes (paralogous genes) genetic losses (complete deletion or presence of a pseudogene) was confirmed using the nucleotide and aminoacid sequences of the reference paralogous gene(s) (*e.g.*, *ureA* gene, to identify the presence of *ureB, ureE, ureF, ureG* and *ureHD*) using blast2seq (https://blast.ncbi.nlm.nih.gov/Blast.cgi). In addition to the previous analyses (BLAST and paired alignments), possible annotation error events were confirmed using the BEACON program (Bacterial Genome Annotation Comparison) (*Kalkatawi, Alam & Bajic, 2015*). Using this

methodology, we identified four types of annotation errors (that do not match what was reported in the gene annotations in the GenBank Flat files), named: assigned as another paralogous gene, assigned as hypothetical protein, gene not reported and no annotation. Error rate was obtained from incorrectly annotated VF genes divided by the total VF detected in this work.

### Ancestral state estimation and phylogenetic analysis

Phylogenetic analysis was conducted on the 17 protein-coding genes (PCGs) encoded for flagella genes (*flaB, flaG, flgI, flgH, flhG1, flhF, fliM, fliL, motA, motB, flgE1, flgD, flgE2, flgK, fliE, flgC* and *flgB*) and two rRNA genes (*rrnL* and *rrnS*) strictly conserved for all *Helicobacter* species and *Campylobacter fetus* (used as out-group). For this selection we removed VF genes with pseudogenes. The dataset of all PCGs and two rRNA genes was concatenated using the Geneious program (*Kearse et al., 2012*). Bayesian inference analysis (BI) was made with Mr Bayes version 3.2 (*Ronquist et al., 2012*) with two MCMC runs each with four chains, run for 1,200,000 generations and a burn-in of 25%. Based on the matrix of presence/absence of VF genes by genome, a manual parsimonious analysis of gene gain and loss was performed on the nodes and taxa of the resulting phylogenetic tree.

### Molecular evolution analysis of virulence factors genes

From the matrix of presence and absence of VF genes, possible gene gain and loss events were identified based on the phylogeny, applying the maximum parsimony method. VF genes that showed drastic changes during the evolution of the *Helicobacter* genus were selected. This selection was based on two criteria: (a) paralogous genes that showed a greater than 80% variation in nucleotide identity from blastn, between closely related species; (b) paralogous genes that showed significant variation in bp size (more than 5%). Once the gene clusters were identified, the coding regions were translated, using the Geneious program (*Kearse et al., 2012*) and multiple alignments were performed with the Muscle program to identify possible insertions or deletions. Blast2seq was performed to identify possible microduplication events or inversions within each gene, in order to detect possible microevolution events.

## RESULTS AND DISCUSSION

### Revision of virulence factors annotation

A total of 945 VF genes were detected in the 22 species of the *Helicobacter* genus and *Campylobacter fetus* (outgroup); most of these belonging to VF classified as flagella genes, representing 80% (756/945) of the total, followed by ureases genes with 10% (94/945). The genome with the highest number of VFs was *Helicobacter pylori* with 89 copies, followed by *Helicobacter acinonychis* and *Helicobacter cetorum* with 55 and 52, respectively. In the *Helicobacter pametensis* genome (NZ_UYIU01000001), no VF genes were detected (Table 1). Likewise, we used blastn program to analyze the *rrnL* and *rrnS* gene sequences of *H. pametensis*. The similarity for both gene sequences with the bacterium *Eikenella exigua* (CP038018; e-value: 0, pairwise identity: 99.3% and e-value: 0,
**Table 1 Gene annotation errors of VF genes in the *Helicobacter* genus.**

| Species | ID | Niche/Host[a,b] | Urease | Adherence | Inmune evasion | Fagellin | Secretion system | Cytotoxins | Plasticity region | Total |
|---|---|---|---|---|---|---|---|---|---|---|
| *Helicobacter pylori* | NC_000915.1 | GHS/Human | 0 (7) | 0 (9) | 0 (5) | 0 (37) | 0 (27) | 0 (1) | 0 (3) | 0 (89) |
| *Helicobacter acinonychis* | NC_008229 | GHS/big cats | 1 (9) | 0 (3) | 0 (5) | 0 (37) | 0 (0) | 0 (1) | 0 (0) | 1 (55) |
| *Helicobacter cetorum* | NC_017735 | GHS/Cetaceans | 3 (9) | 0 (3) | 0 (2) | 0 (37) | 0 (1) | 0 (0) | 0 (0) | 3 (52) |
| *Helicobacter felis* | NC_014810 | GHS/Cat, dog | 3 (9) | 0 (0) | 0 (0) | 2 (35) | 0 (0) | 0 (0) | 0 (0) | 5 (44) |
| *Helicobacter bizzozeronii* | NC_015674 | GHS/Cat, dog | 2 (7) | 0 (0) | 0 (0) | 2 (33) | 0 (0) | 0 (0) | 0 (0) | 4 (40) |
| *Helicobacter suis* | NZ_AP019774 | GHS/Pig | 2 (7) | 0 (0) | 0 (0) | 2 (36) | 0 (0) | 0 (0) | 0 (0) | 4 (43) |
| *Helicobacter heilmanni* | HE984298.2 | GHS/Cat | 6 (11) | 0 (0) | 0 (0) | 6 (36) | 0 (0) | 0 (0) | 0 (0) | 12 (48) |
| *Helicobacter mustelae* | NC_013949 | Gastrointestinal/Ferret | 3 (8) | 0 (0) | 0 (0) | 2 (34) | 0 (0) | 0 (0) | 0 (0) | 5 (42) |
| *Helicobacter canis* | NZ_LR698964 | EHS/Human | 0 (0) | 0 (0) | 0 (0) | 2 (32) | 0 (0) | 0 (3) | 0 (0) | 2 (35) |
| *Helicobacter macacae* | KI669454.1 | EHS/Rhesus macaque | 0 (0) | 0 (0) | 0 (0) | 4 (28) | 0 (0) | 0 (0) | 0 (0) | 4 (28) |
| *Helicobacter fennelliae* | NZ_UGIB01000001 | EHS/Human | 0 (0) | 0 (0) | 0 (0) | 2 (35) | 0 (0) | 0 (0) | 0 (0) | 2 (35) |
| *Helicobacter cinaedi* | NC_020555 | EHS/Human | 0 (0) | 0 (0) | 0 (0) | 3 (36) | 0 (0) | 0 (1) | 0 (0) | 3 (37) |
| *Helicobacter hepaticus* | NC_004917 | EHS/Mice | 0 (7) | 0 (1) | 0 (0) | 2 (35) | 0 (0) | 1 (3) | 0 (0) | 3 (46) |
| *Helicobacter typhlonius* | NZ_LN907858 | EHS/Mice | 0 (0) | 0 (0) | 0 (0) | 2 (36) | 0 (0) | 0 (3) | 0 (0) | 2 (39) |
| *Helicobacter pametensis* | NZ_UYIU01000001 | EHS/Tern | 0 (0) | 0 (0) | 0 (0) | 0 (0) | 0 (0) | 0 (0) | 0 (0) | 0 (0) |
| *Helicobacter bilis* | NZ_CP019645 | EHS/pig, sheep, human, rodent, mice | 2 (7) | 0 (0) | 0 (0) | 4 (30) | 0 (0) | 0 (3) | 0 (0) | 6 (40) |
| *Helicobacter muridarum* | NZ_UGJE01000002 | EHS/Rat, mouse | 2 (7) | 0 (0) | 0 (0) | 4 (36) | 0 (0) | 0 (3) | 0 (0) | 6 (46) |
| *Helicobacter canadensis* | NZ_CM000776 | EHS/Human | 0 (0) | 0 (0) | 0 (0) | 2 (35) | 0 (0) | 0 (0) | 0 (0) | 2 (35) |
| *Helicobacter pullorum* | NZ_LR134509 | EHS/Poultry, Human | 0 (0) | 0 (0) | 0 (0) | 1 (34) | 0 (0) | 0 (1) | 0 (0) | 1 (35) |
| *Helicobacter apodemus* | NZ_CP021886 | EHS/Mice | 2 (6) | 0 (0) | 0 (0) | 1 (33) | 0 (0) | 0 (3) | 0 (0) | 3 (42) |
| *Helicobacter cholecystus* | NZ_LR134518 | EHS/Hamster | 0 (0) | 0 (0) | 0 (0) | 1 (34) | 0 (0) | 0 (3) | 0 (0) | 1 (37) |
| *Helicobacter himalayensis* | NZ_CP014991 | EHS/Marmota | 0 (0) | 0 (0) | 0 (0) | 0 (33) | 0 (0) | 0 (3) | 0 (0) | 0 (36) |
| *Campylobacter fetus* | NC_008599 | EHS/ | 0 (0) | 0 (0) | 0 (0) | 0 (32) | 0 (0) | 0 (8) | 0 (0) | 0 (40) |
| Total | | | 26 (94) | 0 (16) | 0 (12) | 42 (755) | 0 (28) | 1 (36) | 0 (3) | 69 (944) |

**Notes:**

Numbers in parentheses correspond to the total number of VF genes detected in this work.

[a] *Smet et al. (2018)*.

[b] *Hu, Zhu & Lu (2020)*.

pairwise identity: 99.0%; respectively) was observed, thus we choose to remove the sequence from this species during our phylogenetic and molecular evolutionary analyses.

Taking into account all VF genes, 73 were identified as gene annotation errors, meaning an error rate of 7.7%. Of these, flagella genes have the highest level of annotation errors (63.0%), followed by ureases (35.6%) and cytotoxins (1.4%). For VFs classified as: adherence, immune evasion, secretion system and plasticity region; no annotation errors were detected. The species with the highest number of gene annotation errors was *Helicobacter heilmanni* with 17.4% (Table 1 and Table S1). Of the 73 annotation errors in VF genes, 45.2% were assigned as "annotation as another paralogous gene", followed by "incomplete annotation errors" with 52.1%. The others were classified as hypothetical protein assigned genes (single gene) and gene not reported (single gene) (Table S1).

Current sequencing methods produce hundreds of bacterial genomes deposited in databases such as the NCBI database. Nevertheless, genome annotation is a crucial step for the extraction of useful information from genomes. Errors in genic annotation can be generated by several factors. According to *Denton et al. (2014)* a significant number of genomes have a high number of gaps in both coding and non-coding regions or unannotated gene; mainly by incomplete genome assemblies. *Salzberg (2019)* proposes that the major challenges in gene annotation can be divided into two categories: (a) automated annotation of large, but a lot of small gaps in the genomes remains very difficult, and (b) gene annotation errors in draft assemblies lead to errors in annotation that tend to propagate across others species. Thus, the amount of "draft" genomes correlates with errors and this propagation. Due to the increased sequencing of hundreds of genomes and as erroneous annotations are sometimes used as the basis for further genome annotations, resulting in what has been called a "percolation of errors", effect common in mammalian mitochondrial genome (*Prada & Boore, 2019*) and in other reference genomes, supposedly well annotated as chimpanzee, chicken and *Drosophila melanogaster;* among others (*Denton et al., 2014*). Although protein-coding gene detection in prokaryotic genomes is considered a much simpler task than in intron-containing eukaryotic genomes, the number of missing genes in the annotation of prokaryotic genomes is worryingly high (*Warren et al., 2010*). These errors could have a major impact on molecular characterization studies of these VF, as they may represent partial or total deletions in coding regions, generating false deletions in paralogous genes that may not be real, significantly affecting evolutionary analysis in these VFs. Another problem could arise when assigning the presence or absence of a certain VF to a pathogenic phenotype, as in the case of *Helicobacter* species.

## Phylogenetic analyses of *Helicobacter* genus

After confirming the presence of each of the VFs genes, we decided to investigate their presence and analyze the taxonomic distribution of the corresponding genes in the available 22 complete *Helicobacter* genomes. As a first step for this phylogenomic analysis, we established a core-conserved VF genes-based phylogeny on 16 protein-coding genes (PCGs) encoded for flagella genes of *Helicobacter* genus and *Campylobacter fetus*.

In addition, other VF genes conserved between the *Helicobacter* genus and *Campylobacter fetus* species (not used in the phylogeny construction) were also identified. These additional VFs genes conserved are 17 flagella genes (*flaB, flaG, flgI, flgH, flhG1, flhF, fliM, fliL, motA, motB, flgE1, flgD, flgE2, flgK, fliE, flgC* and *flgB*) and two cytotoxins genes (*cdtB* and *cdtC*) present in *Campylobacter fetus* and in the most of *Helicobacter* genomes; since a deletion in one or two genomes is more parsimonious than independent genomic gain in 21 species (Table S1). Therefore, these 33 flagella and two cytotoxins genes conserved to the two genera are represented as the ancestral order A1 (Fig. 1). Thus, the putative ancestral genome of the Campylobacter and Helicobacter genera could contain a conserved core of at least 35 VFs. However, a comparative genomic analysis with more Campylobacter genomes would be necessary to confirm these findings. According to our results, the group composed of *H. apodemus, H. canadensis* and *H. pullorum* would be the

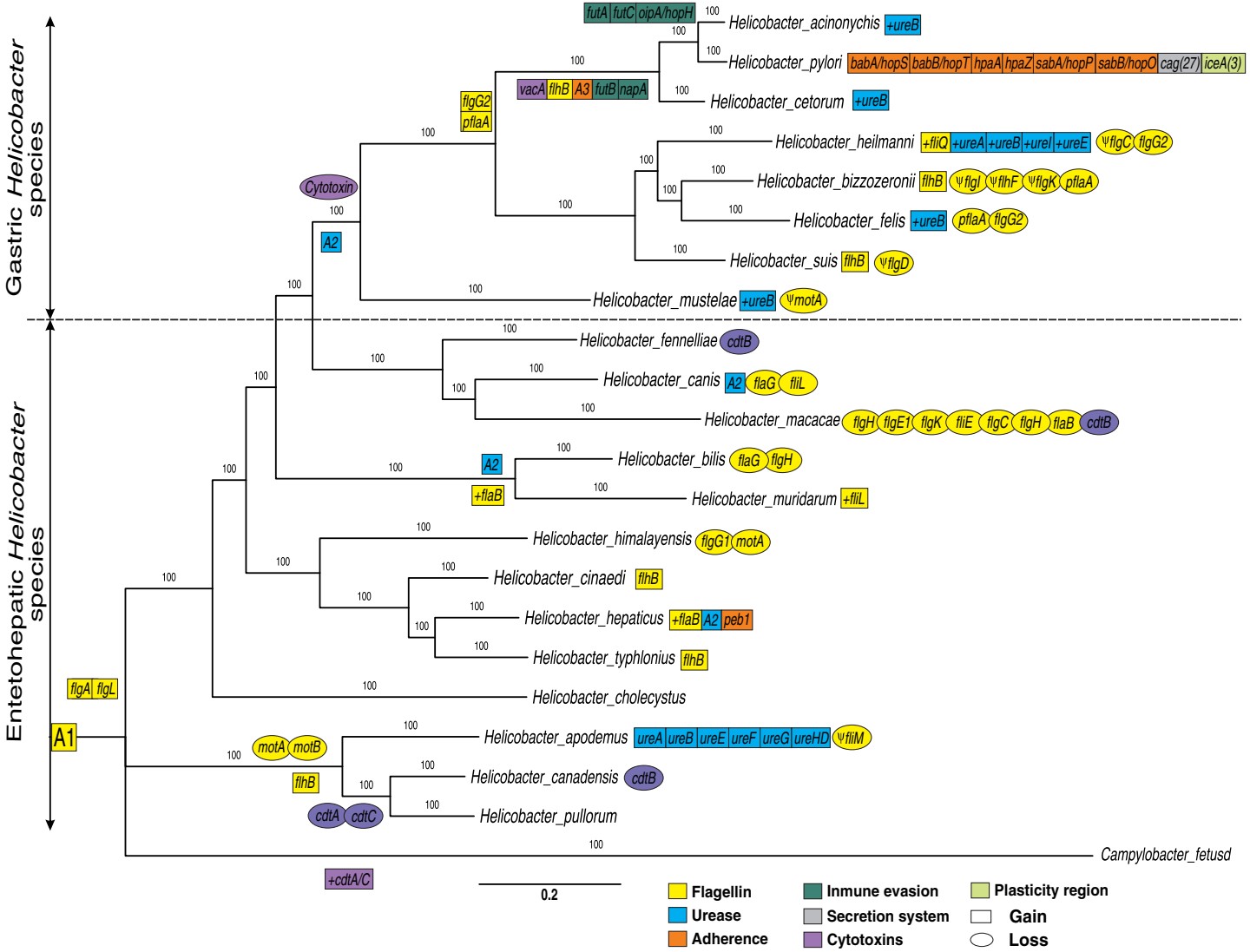

**Figure 1** **Phylogenetic trees of 21 *Helicobacter* species.** A Bayesian inference tree was reconstructed from concatenated nucleotide sequence of 17 protein-coding genes (PCGs) encoded for flagella genes (*flaB, flaG, flgI, flgH, flhG1, flhF, fliM, fliL, motA, motB, flgE1, flgD, flgE2, flgK, fliE, flgC* and *flgB*) and two rRNA genes (*rrnL* and *rrnS*) strictly conserved for all *Helicobacter* species and *Campylobacter fetus* (used as out-group). Ancestral order A1 is represented by the following VF: 17 conserved flagella genes (previously presented), 15 flagella genes (*fliR, fliF, fliG, fliH, flhA, fliA, fliY, fliN, flip, fliD, fliS, flhB_1, fliQ, fliI* and *flgG_2*) and two cytotoxins genes (*cdtB* and *cdtC*) moderately conserved (present in most genomes). Ancestral order A2 is represented by seven ureases genes (*ureA, ureB, ureI, ureE, ureF, ureG* and *ureHD*). Ancestral order A2 is represented by tree adherence genes (*alpA/hopC, alpB/hopB* and *horB*). Rectangles represent gain and ovals represent gene deletions. The plus symbol (+) represents an additional copy of a gene. The Ψ symbol represents a pseudogene.

ancestral clade of the *Helicobacter* genus, with specific genetic gains and losses in each of the three species. Similarly, *flaG* and *flgH* are loss in the *Helicobacter bilis*. Likewise, all EHS are ancestral species and GHS are derived; the latter with an important gain of genetic information in each genome analyzed. For example, the GHS (and some EHS, discussed below) show the relatively conserved presence of ureases (ancestral order A2) and the absence of cytotoxins in this lineage, which would suggest a tissue-specific adaptive process. Therefore, our results show a dynamic between relatively conserved VFs among

the species of the genus, and deletions and/or duplications specific to some lineages or to a certain species that could explain, in part, their pathogenic behavior for each Helicobacter species. Also, our results are consistent with previous phylogenetic analysis (*e.g.*, *Mannion, Shen & Fox, 2018*; *Vinella et al., 2015*; *Cao et al., 2016*); which could elucidate the evolutionary dynamics of these genes. For example, *Mannion, Shen & Fox (2018)* analyzed a total of 62 *Helicobacter* genomes (17 GHS and 45 EHS) indicating that there are significant genetic differences underlying gastric *vs* EHS associated with metabolic adaptation processes and virulence gene potential.

The results will be presented and discussed below for each gene group.

## Flagella genes

In addition to the flagella genes common to both genera, previously presented; our results showed two genes (*flgA*, and *flgL*) that are present only in the *Helicobacter* genus. Pseudogenes for flagella genes were also identified in certain genomes, as in the case of *fliM, flgI, flhF, flgK, flgC, flgD* and *motA*. Similarly, losses and gains of flagella genes were detected in certain clades or *Helicobacter* spp.

*Tsang & Hoover (2015)* show that the bacterial flagellum is a complex nanomachine powered by an ion-driven rotary motor consisting of about 30 different types of proteins whose copy numbers range from a few to thousands. The flagellum is comprised of three basic structures: (a) the basal structure (rotary motor), (b) the hook (universal joint), and (c) the filament (helical propeller) (*Tsang & Hoover, 2015*); organelle that are involved not only in motility and chemotaxis and participate in many additional processes including adhesion, biofilm formation, virulence factor secretion, and modulation of the immune system of eukaryotic cells, contributing to bacterial pathogenicity in *Campylobacter* and *Helicobacter* genera (*Duan et al., 2013*; *Guerry, 2007*; *Ramos, Rumbo & Sirard, 2004*). Previous research has shown that, the *flgA* and *flgL* genes encode hook-associated proteins for the flagellum structure (*Duan et al., 2013*; *Tsang & Hoover, 2015*); therefore, the acquisition of new copies of these two genes could confer a greater pathogenic capacity in these bacteria.

Flagella gene gain in specific GHS clades (except *H. mustelae*), as in the case of *flgG_2* and *pflA* genes were observed. The *flgG* genes are associated with rod formation in the flagellum (*Tsang & Hoover, 2015*), while mutations in *pflA* (paralyzed flagellar protein) gene are also non-motile but possess a flagellum that is found in *Campylobacter jejuni* as defective copy (*Bleumink-Pluym et al., 1999*); possibly associated with an oxidoreductase activity effect in *H. pylori* (*Roszczenko et al., 2012*). Similarly, our analyses indicated the gain of the *flhB* gene in the ancestral genome of *H. acinonychis, H. pylori* and *H. cetorum*. This gene is considered an integral membrane protein in the export dome region, which form an export pore in the *H. pylori* flagellum, of great importance in the function of this organelle (*Ramos, Rumbo & Sirard, 2004*; *Tsang & Hoover, 2015*). Likewise, the presence of pseudogenes in most of the GHS could reinforced the dynamics of gain and loss of flagella genes to different hosts and their pathogenesis process. According to different authors (*Duan et al., 2013*; *Ramos, Rumbo & Sirard, 2004*; *Tsang & Hoover, 2015*), all *Helicobacter*

genomes analyzed (except *H. pametensis*) contain the full complement of genes that would allow the bacterium to assemble a functional flagellum and a secretion system.

## Ureases

Our phylogenetic analyses showed two clearly differentiated clades during the evolution of the *Helicobacter* genus: GHS and EHS (Fig. 1). For example, in GHS a core-conserved of urease genes, called ancestral order A2 (*ureA, ureB, ureI, ureE, ureF, ureG* and *ureH/D*) is observed. However, some EHS as *H. bilis, H. muridarum* and *H. hepaticus*, also presented these ureases (A2). Similarly, the *ureA, ureB, ureE, ureF, ureG* and *ureH/D* genes were detected in *H. apodemus*. Previous studies indicate that urease genes play an important role in stomach colonization by metabolizing urea into ammonia in order neutralize stomach acid needed, which allows the survival and development of the bacteria in the gastric compartment (*Mannion, Shen & Fox, 2018*). In addition, it has been determined that urease can also produce ammonia for nitrogen assimilation instead of acid neutralization in the intestine (pH ~ 6.1) and liver (pH ~ 7.4) (*Ge et al., 2008*; *Mannion, Shen & Fox, 2018*), which would explain the presence of urease gene in specific EHS such as *H. bilis, H. muridarum, H. hepaticus* and *H. apodemus*. Likewise, a gain of extra copies of *ureB* was determined in most of the GHS, as well as the gain of additional copies of ureases (*ureA, ureI* and *ureE*) in *H. heilmanni*, which could be associated to a possible adaptive process in the colonization of the stomach in each specific host. According to *Scott et al. (2000)* urease activity in GHS with pH optima above 7.0 is more than two-fold higher compared to EHS possessing some urease genes. According to previous studies (*Scott et al., 2000*), the presence of *ureI* gene in *H. pylori* is key for acidic pH activation of cytoplasmic urease. In agreement with (*Bury-Moné et al., 2003*), the presence of active aliphatic amidases in GHS is essential for Stomach colonization. Previous studies show that the ability to colonize the gastric mucosa originated more than once in the history of the *Helicobacter* genus, and suggests that acquiring this capacity may be a relatively simple and punctual process, involving a limited number of genes (*Gueneau & Loiseaux-De Goër, 2002*); which makes necessary the acquisition of new copies of paralogous genes as well as other genes that complement the adaptive process in the evolution of these pathogens.

According to *Smet et al. (2018)*, the origin of the common ancestor for the *Helicobacter* genus was 1,52 Mya; where the enteropathic character is the ancestral character and the emergence of colonization to the stomach and subsequent emergence of the ancestral species of all GHS was approximately 610 Kya. This study pointed the presence of an ecological barrier, preventing the genetic exchange between the GHS and EHS, and unraveled many gene flow events within and across species residing in the mammals stomach (*Smet et al., 2018*). Our results would support evidence that this barrier was overcome due to the acquisition of ureases combined with the loss of cytotoxins, which allowed it to survive in a very acidic environment such as the stomach.

## Cytotoxins

Our analyses indicated that cytotoxins (*cdtB* and *cdtC*) are more frequent in *Campylobacter fetus* and EHS, than in GHS in which these Cytotoxins were not found.

Only in the case of *H. pylori* and *H. acinonychis*, we found that they acquired the *vacA* cytotoxin, which only was found in these two bacteria. In addition, some GHS have gained new copies of certain ureases, as in the case of *ureB*, and other ureases like *ureA*, *ureI*, and *ureE* in *H. heilmanni* (Fig. 1).

Moreover, our analyses indicated an ancestor for *Campylobacter* and *Helicobacter* genus (A1) with conserved flagellins and cytotoxins. However, cytolethal distending toxin (CDT) genes (*cdtA*, *cdtB* and *cdtC*) were detected in 10/14 EHS genomes analyzed, but not in any GHS. Previous studies have determined that CDT genes is a bacterial virulence factor produced by several Gram-negative pathogenic bacteria. These genes affect the epithelial cell layer and causes progressive cellular distension and death in several cell lines; and also have been involved in the pathogenicity of the associated bacteria by promoting persistent infection (*Pickett et al., 1996*; *Pons, Vignard & Mirey, 2019*). Likewise, this CDT genes can been promote pro-inflammatory pathology and induce pro-carcinogenic DNA damage in the intestine (*Ge et al., 2008*; *Mannion, Shen & Fox, 2018*). The presence of these genes in EHS and the *Campylobacter* genus would be an ancestral characteristic of these enteropathic bacteria and therefore, the loss of these genes in GHS would be a key step for the colonization of other tissues such as the stomach, which has a more aggressive environment due to its acidity. These results were similar to those previously reported, which could be associated with an adaptive process to a new environment such as the mammalian stomach (*Javed, Skoog & Solnick, 2019*; *Mannion, Shen & Fox, 2018*). In addition to this, the gain of the *vacA* cytotoxin in *H. cetorum*, *H. acinonychis* and *H. pylori* (cytotoxin that produces vacuoles in gastric epithelial cells that results in apoptosis and can trigger inflammation events), would also be associated with a more pathogenic phenotype in comparison with GHS that do not have this gene (*Foegeding et al., 2016*; *Mannion, Shen & Fox, 2018*; *Palframan, Kwok & Gabriel, 2012*).

## Adherence

Adherence VFs, called ancestral order A3 (*alpA/hopC, alpB/hopB, and horB*) were detected in the monophyletic group comprising *H. pylori*, *H. acinonychis* and *H. cetorum* species; with a genetic copy gain of *babA/hopS, babB/hopT, hpaA, hpaZ, peb1, sabA/hopP* and *sabB/hopO* genes in *H. pylori*, unique for this species. However, one copy of *peb1* gene was detected in the *H. hepaticus* genome. Moreover, two Immune evasion genes (*futB* and *napA*) were detected in *H. pylori*, *H. acinonychis* and *H. cetorum* species; while three other genes (*futA, futC* and *oipA/hopH*) were only found in the *H. pylori* and *H. acinonychis* genomes.

According to *Oleastro & Ménard (2013)*, the adherence of *H. pylori* to the mucus layer in gastric epithelium, performs an important role in the initial colonization and persistence of this bacteria in the human stomach during decades or for the entire lifetime; becoming one of the essential bacterial processes to maintain itself in the gastric tissue of the host.

## Immune evasion

We identified the presence of immune evasion VFs in the *H. pylori, H. acinonychis* and *H. cetorum* lineage. These VF play an important role as enzymatic modulates Lewis

antigen glycosylation of *H. pylori* lipopolysaccharides during persistent infection (*Nilsson et al., 2006*). Lewis blood group antigens are present in the gastric mucosa, and the expression of these geneson the bacterial surface may camouflage the bacterium; increasing the survival of *H. pylori* (*Moran, 1996*). Therefore, the presence of these genes confers to these GHS an adaptive advantage, avoiding detection by the host immune system. However, the bacterial persistence has been associated with the presence of other VFs such as immune evasion genes and ureases etc (*Ge et al., 2008*; *Mannion, Shen & Fox, 2018*; *Oleastro & Ménard, 2013*).

## Secretion system and plasticity region

*H. pylori* is the bacterium that has gained the most VF and new copies of paralogous and has uniquely gained 27 Secretion system genes (*cag*) and at least three plasticity region genes (*ice*) during the evolution of the genus (Fig. 1 and Table S1). According to previous studies, the ancient African *H. pylori* strains could be the common ancestor of *H. acinonychis* and *H. ceturum*; presumably by a transfer from humans to other mammals, including pets, as cats and dogs (*Schwarz et al., 2008*; *Smet et al., 2018*). According to this postulate, the jump of *H. pylori* to other mammals would imply the loss of specific genes such as adherence, secretion system and plasticity region genes observed in this species, in order to colonize the stomach of big cats, cetaceans and other domesticated species.

## Molecular evolution

In these analyses, four ureases (*ureA, ureE, ureF* and *ureH/D*) and two flagella genes (*flgA* and *flgE*) were identified as the VFs that varied the most during the evolution of the *Helicobacter* genus. In the monophyletic group composed of *H. pylori, H. acinonychis* and *H. cetorum* species, these four ureases were conserved for more than 80% of their amino acid sequence identity. In the other GHS and the two EHS (*H. bilis, H. muridarum* and H. *hepaticus*), the amino acid identity of the ureases decayed by less than 70% (Fig. 2).

For example, in *ureA* gene, the amino acid identity in *H. pylori, H. acinonychis* and *H. cetorum* species is higher than 94%, in the other species it is lower than 76% (Fig. 2A). In the first three species, the gene codifies a protein with 239 aa, whereas in the other species, this protein has a length between 235 and 193 aa. The protein alignment showed that the end of the sequence had the lowest protein identity (about 50%), where a gain of 13 aa at the end of the gene was observed in the *H. pylori, H. acinonychis* and *H. cetorum*. In *ureE*, the amino acid identity in *H. pylori, H. acinonychis* and *H. cetorum* species is greater than 87%, in the other species it is less than 47% (Fig. 2B). In the first three species, the protein has a length of 171 aa, whereas in the other species, this protein has a length between 171 and 187 aa, which would indicate a progressive loss of genetic information. In *ureF*, the amino acid identity in *H. pylori*, *H. acinonychis* and *H. cetorum* species is greater than 81%, in the other species it is less than 64% (Fig. 2C). In the first three species, the gene product has a length of 255, 250 and 245 aa, respectively, while in the other species this protein has a length between 247 and 221 aa, which would indicate a progressive gain of genetic information through the evolution of these species. Finally, in *ureH/D*, the amino acid identity in *H. pylori, H. acinonychis* and *H. cetorum* species is

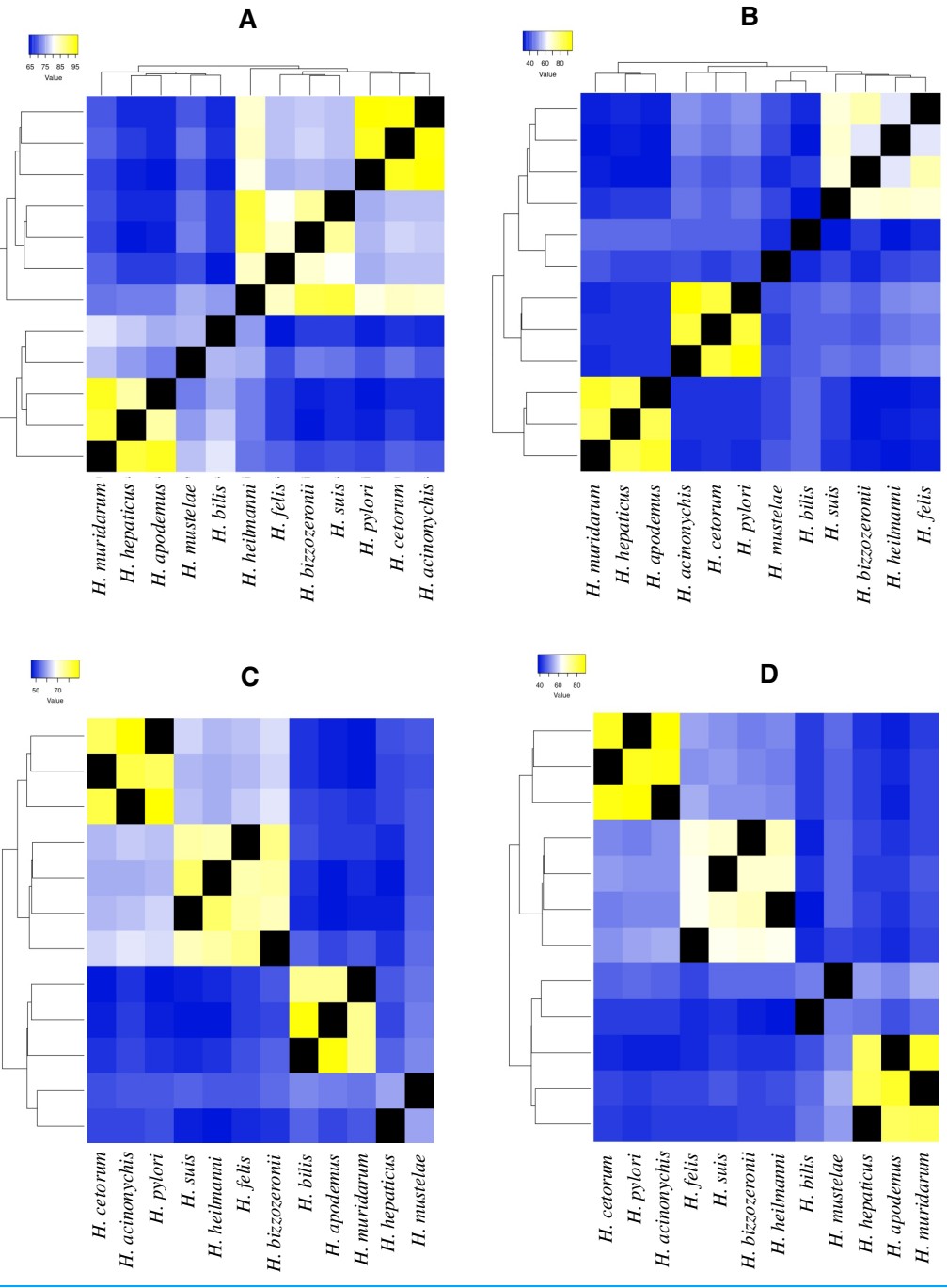

**Figure 2 Heatmap of amino acid identity among four ureases genes of higher genetic variability.**
(A) Amino acid identity analyses for *ureA* gene in 12 *Helicobacter* species. (B) Amino acid identity analyses for *ureE* gene in 12 *Helicobacter* species. (C) Amino acid identity analyses for *ureF* gene in 12 *Helicobacter* species. (D) Amino acid identity analyses for *ureH/D* gene in 12 *Helicobacter* species.

greater than 85%, in the other species it is less than 54% (Fig. 2D). In the first three species, the gene has a length of 266, 264 and 264 aa, respectively, similar to the length observed in other species of the genus.

We found similar results for the two flagella genes (*flgA* and *flgE_2* gene), which the amino acid identity in *H. pylori*, *H. acinonychis* and *H. cetorum* species is greater than 80%, in the other species it is less than 45% (Fig. S1). In these genes, the greatest difference in protein length was detected in the *flgA* gene, which the gene in *H. pylori* codifies a protein with 219 aa, compared with that observed for this protein in other *Helicobacter* species, whose length ranged between 222 and 320 aa.

Urease is very abundant in *H. pylori*, accounting for 10% of total soluble proteins and its maturation requires four accessory proteins for nickel delivery into the active site (*UreE-F-G-H*) (*Carter et al., 2009*). These genes play an essential role in stomach colonization by metabolizing urea into ammonia in order neutralize stomach acid need, then is central to the pathogenesis of *H. pylori* infection and disease (*Šterbenc et al., 2019*; *Mannion, Shen & Fox, 2018*; *Vinella et al., 2015*). As expected, urea uptake and urease genes were identified in all GHS, although unexpectedly in some EHS which could facilitate survival in the acidic gastrointestinal environment during transmission to new hosts (*Mannion, Shen & Fox, 2018*).

According to *Mannion, Shen & Fox (2018)*, some virulence genes are shared among all *Helicobacter* spp., while other contribute to the unique virulence profiles that differentiate gastric and EHS from each other. Our analysis of selected urease genes (*ureA, ureE, ureF* and *ureH/D*) revealed a greatest variation in amino acid composition, indicating strong evidence of accelerated microevolution in the *H. pylori*, *H. acinonychis* and *H. cetorum* lineage compared to other GHS; this would corroborate the hypothesis presented above. Likewise, this same behavior in the four urease genes, as well as in the two flagella genes evaluated (*flgA* and *flgE*) could be exist in the mechanisms by which GHS and EHS are capable of promoting a pathogenic infection in their respective hosts.

## CONCLUSIONS

In sum, our current findings pointed that an accurate annotation of all protein-coding sequences (CDSs) of VFs is useful to exploit the rapidly growing repertoire of completely sequenced prokaryotic genomes of special pathogenic impact, *e.g.*, *Helicobacter* genus. Crucially, this study provided evidence that supports a process of molecular adaptation from EHS to GHS to colonize and persist in the stomach of mammals, showing gene-specific gains and losses during the evolution of the *Helicobacter* genus. Also, the findings from this study provide new evidence of the soft relationship among VF copy variability, adaptive process and pathogenic phenotype in this group of bacteria.

### Funding

This work was supported by Universidad del Tolima, Colombia and Universidade Federal de Sao Carlos, Brasil. Carlos F Quiroga was supported by Oficina de Investigaciones y Desarrollo Científico de la Universidad del Tolima by postdoctoral fellowships (4/2019). We also received financial support *via* the grant #2014/0690-4, Sao Paulo Research

Foundation (FAPESP). The funders had no role in study design, data collection and analysis, decision to publish, or preparation of the manuscript.

## Grant Disclosures

The following grant information was disclosed by the authors:
Universidad del Tolima, Colombia and Universidade Federal de Sao Carlos, Brasil.
Oficina de Investigaciones y Desarrollo Científico de la Universidad del Tolima by Postdoctoral Fellowships: 4/2019.
Sao Paulo Research Foundation (FAPESP): #2014/0690-4.

## Competing Interests

The authors declare that they have no competing interests.

## Author Contributions

- Carlos F. Prada conceived and designed the experiments, performed the experiments, analyzed the data, prepared figures and/or tables, authored or reviewed drafts of the article, and approved the final draft.
- Maria A. Casadiego performed the experiments, prepared figures and/or tables, and approved the final draft.
- Caio C. M. Freire conceived and designed the experiments, analyzed the data, authored or reviewed drafts of the article, and approved the final draft.

## Data Availability

Our work was based on public available genomes, which accession numbers were listed in the manuscript.

## Supplemental Information

Supplemental information for this article can be found online at http://dx.doi.org/10.7717/peerj.13120#supplemental-information.

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
