# Peer review of "Evolution of Helicobacter spp: variability of virulence factors and their relationship to pathogenicity"

_PeerJ, doi:10.7717/peerj.13120_

## Round 0.1 · original submission · Major Revisions

I must say that the critical comments of the second reviewer are very serious and without full answers to them in the revised version I don't see any sense to publish this research. I would strengthen that evolutionary and statistical analyses are not satisfactory.

·

Basic reporting

The manuscript of Quiroga et al made a comparative analysis of putative virulence factors of 22 Helicobacter species from which the whole genome sequence is available. The authors wrote a clear and direct paper. The manuscript provided enough background and the scientific rationale for the specific aims. The manuscript has the structure of any other scientific paper and included tables and figures in addition of supplemental data. The list of references is complete and the authors described the results as they were obtained.

Experimental design

The experimental design since appropriate but as the authors recognized there is a clear difference between Gastric and Enteric Helicobacter and the phylogenetic tree confirmed that presumption.
The reported of lack of accurate annotation is relevant but also the authors use some broad definition of virulence factors to survival, colonization and classical virulence factors such as CagA and VacA.
Probably is outside of the scope of this paperboy I wish the authors tested the presence of the 89 virulence factors reported in 26695 H. pylori against many of the others H. pylori strains from which the whole genome sequence is available. The paper is solid

Validity of the findings

I feel that the authors conclusions are well stated a d were based on the results obtained. Moreover the paper is original and has a novelty that no other paper has provided and I feel that can be of interest for your readership

Reviewer 2 ·

Basic reporting

This paper investigated the ancestry of virulence factor in Helicobacter genus by the gene and loss analysis, it can fill the knowledge gap that flagella and adherence genes are important in the adaptation to different species. However, there are several matters that should be clarify.

Experimental design

1. In the previous study analyzing the virulence of H. pylori genus, the reference used Campylobacter jejuni (L. Cover, Cao). Could you explain specific reason of the use of Campylobacter fetus as the reference?
2. Please add technical definition of what is “gain and loss” (on what based, such as how many similarity percentage etc ) in the method section.

Validity of the findings

3. In the result section (page 8 Line 162) it was reported that the error was about 7.7%, but there was no description on how the author identified these error, please add briefly in the method section.
4. The title of this section is “Revision of Virulence Factors Annotation” but the explanation on how these error “revised” was unclear.
5. In page 9 line 76, the the error was mainly due to the incomplete genome assemblies , or so called draft genome. It seems to be contradictory with the sentence in method section ;page 5 line 89, “ these analysis just use the complete genomes so it does not seems to relate. Please clarify this.
6. In these part “In addition, other VF genes common .. “(page 10 line 205), there were others commonly found genes. But flaG and flgH are loss in the Helicobacter bilis. The word “common” might be interpreted as 100% presence in all genes. Please clarify and add more clear explanation because this is the most important result.
7. It might be easier to see if the authors can summarize these 33 genes into a table and define how much gain and loss occurred in each species.
8. The title of section “Phylogenetic analyses of Helicobacter genus” did not seems to match with the content. The reader would expect to see the phylogenetic tree figure and the interpretation. Please elaborate more line 209 “represented as ancestral order”. Please add how the author can conclude which one is closer to the ancestor.
9. It is confusing if the phylogenetic tree was constructed using 17 PCGs (page 6 line 124) but why the Figure 1 ancestral order were represented by 7 urease genes. Can you describe what is ancestral order?We also cannot see A2 in the figure.
10. Please describe in the results what is A1 and what is A2.
11. In the Phylogenetic analysis section (line 197), please compare phylogenetic analysis result in this study with the phylogenetic analysis from previous studies for the discussion (the authors mentioned several previous study in the introduction).
12. Page 17, Molecular evolution. It seems interesting and clear that the authors presented the microevolution in the heatmap (Figure 2). However, the figure for the flgA and flgE genes (as mentioned in page 18 line 379 are missing).

Additional comments

13. There are typos in many parts of the manuscript, such as line 178, 181 etc..

---

## Round 0.2 · Major Revisions

There were 12 critical points in the previous review. I cannot consider your rebuttal as satisfactory. You must answer all questions. Moreover, I expect that in your revised version a reader can find real validation of your findings.

---

## Round 0.3 · accepted · Accept

The authors carefully changed the text considering all critical comments of the reviewers.